# Analysing the Linearity of Linguistic Relations in Language Model Embedding Spaces

Vasudevan Nedumpozhimana
ADAPT Research Centre
Trinity College Dublin, Ireland
vnedumpo@tcd.ie

Fathima Thekkekara
Indian Institute of Technology Bombay
Mumbai, India
fathima@iitb.ac.in

John Kelleher
ADAPT Research Centre
Trinity College Dublin, Ireland
john.kelleher@tcd.ie

## Abstract

We propose a framework to analyse how strongly different linguistic relations are linearly encoded in language model embedding spaces. We formalise linear encoding via a constrained linear approximation over related and unrelated word pairs and apply this to an extended BATS dataset covering inflectional, derivational, lexicographic, and encyclopedic relations in GloVe, RoBERTa, and ModernBERT. Our experiments show near-perfect linear encodings for inflectional and derivational relations, but substantially higher errors for lexicographic and encyclopedic relations, especially for one-to-many and many-to-many associations. We also find that RoBERTa and ModernBERT generally encode relations more linearly than GloVe. These results indicate that our framework can reveal which relational structures are most linearly accessible in embeddings, offering a compact tool for probing and comparing relational geometry across models.

## 1 Introduction

Large language models (LLMs) and other deep learning–based natural language processing models function by transforming the input text into high-dimensional numerical vectors called embeddings, in which meaning is represented in a distributed way. What such an embedding vector represents depends on its numerical values and its position relative to other embedding vectors in the embedding space. This distributed, high-dimensional coding makes language processing models powerful but also opaque, because it is hard to see what information—especially about linguistic relationships—is encoded where, and how it influences model behaviour.

Probing methods are a widely used way to study what kinds of information are encoded in a deep learning based language processing model's internal representations by testing what information can be recovered from these representations (Conneau et al., 2018; Nedumpozhimana & Kelleher, 2024). This work proposes a novel framework to analyse the latent representations of language processing models, and goes beyond the standard probing methods in two key ways. First, rather than simply testing whether particular information is present in an embedding, our framework can be used to understand how it is encoded in the embedding space, and more specifically, whether it is represented in a linear or non-linear form. The theoretical inspiration for our approach is the linear representation hypothesis (Park et al., 2024), which proposes that, for at least some linguistic properties, models organise their internal space linearly. Such linearly encoded linguistic properties may be more accessible to the model's downstream computation as compared to other information in the model's representations, and so may disproportionately influence the model's behaviour. Thus, by identifying which information is encoded linearly, we can begin to

explain which information strongly drives model behaviour and how we might safely intervene on this behaviour. Second, while much existing work has applied probing to individual concepts or token-level properties, we focus on linguistic relations (such as syntactic or semantic relations). Adopting a relation-based rather than concept-based perspective is both novel and advantageous because a relational view asks how models represent the links between elements in text, which drive many downstream behaviours. The proposed framework is defined for arbitrary linguistic relations (word-to-word, word-to-sentence, and sentence-to-sentence), and can handle varying relational complexity (one-to-one, one-to-many, and many-to-many).

## 2   Linearly Encoded Relations

We formalise the concept of the linearity of a relation by defining that any relation $r$ is linearly encoded in the embedding space if there exist two linear operators that map representations of a pair to the same embedding vector if and only if that pair is related. In linear algebra, a linear relation is one where related pairs $(e_1, e_2)$ in a module $M$ over a ring $R$ satisfy a linear equation $f_1 e_1 + f_2 e_2 = 0$, where $f_1$ and $f_2$ are two elements in the ring $R$ (Lang, 2002). In our case, we consider the embedding space as a Module of all $d$-dimensional vectors ($\mathbb{R}^d$) over the ring of all $d \times d$ square matrices ($\mathbb{R}^{d \times d}$). Note that the space of square matrices over matrix addition and matrix multiplication is a ring, and therefore, the set of all $d$-dimensional vectors over $d \times d$ matrices is a module.

Suppose $r$ is the target linguistic relation, and $t_1$ and $t_2$ are related linguistic expressions (i.e., $(t_1, t_2) \in r$). Let $E$ be the embedding mapping that maps any linguistic expression to a $d$-dimensional embedding vector in the embedding space ($\mathbb{R}^d$) and let $e_1$ and $e_2$ be the two $d$-dimensional embedding vectors of linguistic expressions $t_1$ and $t_2$ represented as column matrices. Now, if the relation $r$ is linearly encoded in the embedding space, then there exist two $d \times d$ square matrices $\boldsymbol{L}_r$ and $\boldsymbol{R}_r$ that correspond to the relation $r$ that maps both $e_1$ and $e_2$ to the same vector. To align this definition with the standard definition of a linear relation, we can multiply the $\boldsymbol{R}_r$ operator matrix by $-1$, so that it will obey the linear equation $\boldsymbol{L}_r e_1 + \boldsymbol{R}_r e_2 = 0$. For more notational simplicity, we can concatenate $e_1$ and $e_2$ to create a single $2d$-dimensional vector $e_{12}$, and column wise concatenate $\boldsymbol{L}_r$ and $\boldsymbol{R}_r$ to create a single $d \times 2d$ matrix $\boldsymbol{M}_r$. Then we can formally define that if a relation $r$ is linearly encoded in the embedding space defined by the embedding mapping $E$, then there exists an $\boldsymbol{M}_r$, such that:

$$\boldsymbol{M}_r e_{12} = 0 \iff (t_1, t_2) \in r \tag{1}$$

## 3   Linear Approximation

Since many linguistic relations will not satisfy the exact linear encoding condition in 1, we next define a linear approximation that quantifies how closely a relation can be represented linearly. Some relations can be approximately encoded linearly; when the embeddings contain noise, this can often be corrected with slight modifications, whereas some relations can only be linearly encoded by excluding extreme instances. Even for relations that are not exactly linearly encodable, it can still be informative to quantify the degree of linearity they exhibit.

One can observe that if unrelated pairs of expressions are not considered, a trivial solution (i.e., $\boldsymbol{M}_r = 0$) exists for any relation. To avoid this, it is necessary to include unrelated pairs in addition to related ones. For related pairs, according to the condition 1, $\boldsymbol{M}_r e_{12}$ should be the zero vector, and hence its Euclidean norm is 0. In contrast, for unrelated pairs $(\bar{t_1}, \bar{t_2})$, $\boldsymbol{M}_r \bar{e}_{12}$ should not be the 0 vector (where $\bar{e}_{12}$ denotes the concatenated embedding of $\bar{t_1}$ and $\bar{t_2}$), and therefore its Euclidean norm is strictly greater than 0. In this case, by appropriately scaling $\boldsymbol{M}_r$, we can ensure that the Euclidean norm is greater than or equal to 1 without affecting the related pairs. Therefore, we rewrite the condition 1 as:

$$\|\boldsymbol{M}_r e_{12}\|^2 = 0, \forall (t_1, t_2) \in r \text{ and } \|\boldsymbol{M}_r \bar{e}_{12}\|^2 \geq 1, \forall (\bar{t_1}, \bar{t_2}) \notin r \tag{2}$$

Based on this condition, we define the linear approximation of a linguistic relation $r$ as the matrix $\tilde{\boldsymbol{M}}_r$ such that,

$$\tilde{\boldsymbol{M}}_r = \arg\min_{\boldsymbol{M}} \Big\{ \sum_{(t_1,t_2)\in r} \|\boldsymbol{M}\boldsymbol{e}_{12}\|^2 \Big\} \text{ such that, } \|\boldsymbol{M}\bar{\boldsymbol{e}}_{12}\|^2 \geq 1, \forall(\bar{t_1},\bar{t_2}) \notin r \qquad (3)$$

Practically, it is not possible to consider all related pairs and all unrelated pairs, and therefore, we select $p$ related pairs and $n$ unrelated pairs. We can concatenate the embeddings of $p$ related pairs to create a $2d \times p$ matrix $\boldsymbol{P}_r$, and $n$ unrelated pairs to create a $2d \times n$ matrix $\boldsymbol{N}_r$. Now, we can restate the above optimisation problem as,

$$\tilde{\boldsymbol{M}}_r = \arg\min_{\boldsymbol{M}} \big\{ tr((\boldsymbol{M}\boldsymbol{P}_r)^T(\boldsymbol{M}\boldsymbol{P}_r)) \big\} \text{ such that, } \|\boldsymbol{M}\boldsymbol{v}\|^2 \geq 1, \forall \boldsymbol{v} \in columns(\boldsymbol{N}_r) \qquad (4)$$

This optimisation can be further simplified and formulated as a linear programming problem:

$$\tilde{\boldsymbol{x}} = \arg\min_{x}\{\boldsymbol{c}.\boldsymbol{x}\} \text{ such that, } \boldsymbol{A}\boldsymbol{x} \geq 1, \boldsymbol{x} \geq 0 \qquad (5)$$

Where $\boldsymbol{A} = (\boldsymbol{N}_r^T\boldsymbol{U}) \odot (\boldsymbol{N}_r^T\boldsymbol{U})$, $\boldsymbol{U}$ is the left-singular matrix of $\boldsymbol{P}_r$, $\odot$ is the element-wise multiplication, and $\boldsymbol{c}$ is the element-wise square of singular values of $\boldsymbol{P}_r$. (See Appendix A for more details.) The optimum $\boldsymbol{x}$ (i.e., $\tilde{\boldsymbol{x}}$) for a relation $r$ from this formulation serves two roles: its objective value $c \cdot \tilde{x}$, normalised by the number of related pairs, defines the approximation error for relation $r$ (see Section 4), and its components determine the linear operator $\tilde{\boldsymbol{M}}_r = diag(\sqrt{\tilde{\boldsymbol{x}}})\boldsymbol{U}^T$ where $\sqrt{\tilde{\boldsymbol{x}}}$ is the element-wise square root of $\tilde{\boldsymbol{x}}$.

## 4 Are relations encoded linearly in representational spaces?

We conducted a preliminary empirical analysis to check whether the proposed framework can be used to investigate whether linguistic relations are linearly encoded in the representational space of some well-known language processing models.

For this experiment, we extended the BATS dataset (Gladkova et al., 2016), which contains 40 word-to-word relations, including 10 Inflectional relations, 10 Derivational relations, 10 Lexicographic relations, and 10 Encyclopedic relations. For each of these 40 relations, the BATS dataset lists 50 pairs of words for which that relation holds. To create a dataset for our experiments for each relation in BATS, we manually created 50 more related pairs, and created an extended BATS dataset with 100 related pairs for each relation. Many relations we consider in this experiment are one-to-many or many-to-many, and in such cases, every word is paired individually with every other related word, and hence, the number of data points varies for different relations. Details of the number of related pairs are shown in Table 1.

For each of these relations, we also created a set of unrelated pairs by using the words already present in the dataset. In this process of creating unrelated pairs, we treated the domain (set of all words that come first in the related pairs) and the range (set of all words that come second in the related pairs) as two different categories. By this segragation we avoid assuming that the domain and range of a relation should be the same. We then created the full Cartesian product of domain and range, and treated any pair not in the related set as an unrelated pair. As in the case of related pairs, the number of unrelated pairs also varies from one relation to another relation, and these details are shown in Table 1.

To generate representations of words, we used three models: a non-neural representation model, GloVe (Pennington et al., 2014); a neural representation model, RoBERTa (Liu et al., 2019); and one of the most recent neural representation models, ModernBERT (Warner et al., 2025). While generating the GloVe representation, if the word is not in the vocabulary, we randomly assign a fixed 300-dimensional representation for such words. To generate RoBERTa and ModernBERT representations, we selected the average final layer token embeddings (note that a word can have multiple tokens) generated by the model from the input word. Then we analysed whether relations are linearly encoded in these representational spaces by linearly approximating these relations and calculating the error of approximation. The approximation error for a relation is the objective function (5) normalised by the number of related pairs; zero error implies perfect linear encoding.

Table 1: Statistics of dataset and linear approximation errors (macro average) of BATS relations.

| Relations | # Related pairs | | | # Unrelated pairs | | | Avg error of approximation | | |
|---|---|---|---|---|---|---|---|---|---|
| | Min | Max | Avg | Min | Max | Avg | GloVe | RoBERTa | ModernBERT |
| Inflectional | 100 | 131 | 108.6 | 9900 | 17030 | 11765.2 | 0 | 0 | 0 |
| Derivational | 101 | 202 | 124.7 | 9999 | 20200 | 13909.8 | 0 | 0 | 0 |
| Encyclopedic | 104 | 284 | 177.2 | 2751 | 11539 | 7544.9 | 0.4704 | 0.4685 | 0.4549 |
| Lexicographic | 209 | 1988 | 965.1 | 12814 | 197900 | 63923.3 | 0.9201 | 0.8139 | 0.8360 |

From our empirical analysis, we found that both Inflectional and Derivational relations are linearly encoded in the representational spaces of all three models (with 0 approximation error). However, for Lexicographic and Encyclopedic relations, none of the models has a perfect linear encoding. We also found that, although the average values of errors of linear approximations are comparable for all three models, the RoBERTa and ModernBERT average scores are better than the GloVe average score. This shows that in more recent and powerful language models, relations are encoded more linearly. However, when we compare RoBERTa with ModernBERT, RoBERTa is better on Lexicography relations, and ModernBERT is better on Encyclopedic relations.

Generally, we found that relations that are one-to-one are more likely to encode linearly in representational space. For example, all inflectional and derivational relations (morphological relations) are one-to-one, and we found near-perfect linear approximations for these relations. However, for relations with one-to-many or many-to-many related pairs, i.e., Lexicographic and Encyclopedic semantic relations, we observed that it is harder to find a linear approximation. For example, for one of the lexical relations, 'part-whole', which is a many-to-many relation, we got an above 1 average error of approximation for all three representation models (GloVe: 1.3017, RoBERTa: 1.1070, and ModernBERT: 1.1690). However, for the lexical relation with relatively fewer many-to-many related pairs, 'antonyms-binary', we got lower approximation errors (GloVe: 0.3309, RoBERTa: 0.3246, and ModernBERT: 0.3257). We observed a similar pattern in Encyclopedic relations.

When we further analysed the Encyclopedic relations, we found that relations that are non-deterministic or non-exclusive (one-to-many) are hard to approximate linearly compared to relations with strong, nearly one-to-one associations between entities. For example, in the case of 'country-language', languages like Malayalam and Hindi have a strong association with India and are therefore more linearly encoded in the embedding space than English, whose association with India is diffuse and non-exclusive. Similarly, for the 'thing–colour' relation, many related pairs such as 'banana' and 'green' are context-dependent/non-deterministic because a 'banana' can be 'green', but it can also be 'yellow', and for these pairs, we obtained higher approximation errors.

## 5   Conclusion

In this work, we proposed a framework for analysing the linearity of linguistic relations and applied it to 40 word-to-word relations of varying complexity in GloVe, RoBERTa, and ModernBERT. We found that inflectional and derivational relations admit near-perfect linear encodings, whereas lexicographic and encyclopedic relations—especially one-to-many and many-to-many mappings—yield substantially higher approximation errors, with RoBERTa and ModernBERT generally encoding relations more linearly than GloVe. This shows that our framework can pinpoint which relational structures are most linearly accessible in current language models and provides a practical tool for comparing relational geometry across architectures.

Acknowledgments

This work was partly supported by the ADAPT Centre which is funded under the SFI Research Centres Programme (Grant 13/RC/2106_P2) and is co-funded under the European Regional Development Funds.

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

## A   LP Formulation for Linear Approximation

The optimisation stated in 4 for linear approximation has a quadratic objective function with $2d^2$ variables and $n$ quadratic constraints. To simplify this optimisation, we apply singular value decomposition on $\boldsymbol{P}_r$ ($2d \times p$ matrix created by concatenating embeddings of $p$ related pairs), such that $\boldsymbol{P}_r = \boldsymbol{U}\Sigma\boldsymbol{V}^T$. We assume the SVD of $\boldsymbol{M}$ is $\boldsymbol{Q}\boldsymbol{S}\boldsymbol{R}^T$, then we constrain our search space of $\boldsymbol{M}$ such that the right singular matrix of $\boldsymbol{M}$ is the same as the left singular matrix of $\boldsymbol{P}_r$ (i.e., $\boldsymbol{R} = \boldsymbol{U}$). Then we can rewrite the objective function of the above optimisation problem as,

$$tr((\boldsymbol{M}\boldsymbol{P}_r)^T(\boldsymbol{M}\boldsymbol{P}_r)) = tr((\boldsymbol{Q}\boldsymbol{S}\boldsymbol{U}^T\boldsymbol{U}\Sigma\boldsymbol{V}^T)^T(\boldsymbol{Q}\boldsymbol{S}\boldsymbol{U}^T\boldsymbol{U}\Sigma\boldsymbol{V}^T)) \tag{6}$$

By using the unitary property of singular matrices, this can be simplified further.

$$
\begin{aligned}
tr((\boldsymbol{M}\boldsymbol{P}_r)^T(\boldsymbol{M}\boldsymbol{P}_r)) &= tr((\boldsymbol{Q}\boldsymbol{S}\Sigma\boldsymbol{V}^T)^T(\boldsymbol{Q}\boldsymbol{S}\Sigma\boldsymbol{V}^T)) & (7) \\
&= tr(\boldsymbol{V}\Sigma^T\boldsymbol{S}^T\boldsymbol{Q}^T\boldsymbol{Q}\boldsymbol{S}\Sigma\boldsymbol{V}^T) & (8) \\
&= tr(\boldsymbol{V}\Sigma^T\boldsymbol{S}^T\boldsymbol{S}\Sigma\boldsymbol{V}^T) & (9)
\end{aligned}
$$

Here, the $\Sigma$ will be a $2d \times p$ matrix and $\boldsymbol{S}$ will be a $d \times 2d$ matrix, and therefore the number of non-zero diagonal entries of $\Sigma$ will be at most $2d$, and that of $\boldsymbol{S}$ will be at most $d$. Let the diagonal entries of $\Sigma$ be $\sigma_1, \sigma_2, \ldots \sigma_{2d}$ and the diagonal entries of $\boldsymbol{S}$ be $s_1, s_2 \ldots s_d$, then $\Sigma^T \boldsymbol{S}^T \boldsymbol{S} \Sigma$ will be a diagonal matrix with entries $\sigma_1^2 s_1^2, \sigma_2^2 s_2^2, \ldots \sigma_d^2 s_d^2$. The trace of a matrix is the sum of its singular values; therefore, the $tr(\boldsymbol{V} \Sigma^T \boldsymbol{S}^T \boldsymbol{S} \Sigma \boldsymbol{V}^T)$ will be $\sigma_1^2 s_1^2 + \sigma_2^2 s_2^2 + \cdots + \sigma_d^2 s_d^2$. Let $\boldsymbol{x} = (s_1^2, s_2^2, \ldots s_d^2)$ and $\boldsymbol{c} = (\sigma_1^2, \sigma_2^2, \ldots \sigma_d^2)$ represented as column matrices. Then we can rewrite 9 in terms of $\boldsymbol{x}$ and $\boldsymbol{c}$ as,

$$tr((\boldsymbol{M}\boldsymbol{P}_r)^T(\boldsymbol{M}\boldsymbol{P}_r)) = \boldsymbol{c}^T\boldsymbol{x} \tag{10}$$

Similarly, the constraints of the optimisation problem can be rewritten as,

$$\|\boldsymbol{M}\boldsymbol{v}\|^2 \geq 1 \iff (\boldsymbol{Q}\boldsymbol{S}\boldsymbol{U}^T\boldsymbol{v})^T(\boldsymbol{Q}\boldsymbol{S}\boldsymbol{U}^T\boldsymbol{v}) \geq 1 \tag{11}$$

$$\iff \boldsymbol{v}^T\boldsymbol{U}\boldsymbol{S}^T\boldsymbol{Q}^T\boldsymbol{Q}\boldsymbol{S}\boldsymbol{U}^T\boldsymbol{v} \geq 1 \tag{12}$$

$$\iff \boldsymbol{v}^T\boldsymbol{U}\boldsymbol{S}^T\boldsymbol{S}\boldsymbol{U}^T\boldsymbol{v} \geq 1 \tag{13}$$

$$\iff ((\boldsymbol{v}^T\boldsymbol{U}) \odot (\boldsymbol{v}^T\boldsymbol{U}))\boldsymbol{x} \geq 1 \tag{14}$$

Where $\odot$ is the element-wise multiplication. Now we can rewrite the optimisation in terms of $\boldsymbol{x}$,

$$\tilde{\boldsymbol{x}} = \arg\min_{\boldsymbol{x}}\{\boldsymbol{c}^T\boldsymbol{x}\} \text{ such that,} \tag{15}$$

$$\boldsymbol{A}\boldsymbol{x} \geq \boldsymbol{1}, \boldsymbol{x} \geq 0 \tag{16}$$

Where $\boldsymbol{A} = (\boldsymbol{N}_r^T\boldsymbol{U}) \odot (\boldsymbol{N}_r^T\boldsymbol{U})$ and $\boldsymbol{1}$ is a unit column matrix.

Finally, from the solution of the linear programming problem (i.e. $\tilde{\boldsymbol{x}}$), we can reconstruct the approximate linear operator matrix $\tilde{\boldsymbol{M}}_r$. From our formulation, the singular value decomposition of $\tilde{\boldsymbol{M}}_r$ is $\boldsymbol{Q}\boldsymbol{S}\boldsymbol{U}^T$. Here, the left singular matrix $\boldsymbol{Q}$ will cancel out in the optimisation, and therefore we can set it as the identity matrix without affecting the approximation error. We already assumed that the right singular matrix $\boldsymbol{R}$ is the same as $\boldsymbol{U}$. The optimum singular value matrix, $\boldsymbol{S}$, can be estimated as $diag(\sqrt{\tilde{\boldsymbol{x}}})$, where $\sqrt{\tilde{\boldsymbol{x}}}$ is the element-wise square root of $\tilde{\boldsymbol{x}}$. Then, by combining these, we can write:

$$\tilde{\boldsymbol{M}}_r = diag(\sqrt{\tilde{\boldsymbol{x}}})\boldsymbol{U}^T$$

Furthermore, the approximation error—i.e., the minimum value in 4—is equal to $\boldsymbol{c}^T\tilde{\boldsymbol{x}}$.

