# OpenReview forum: "Analysing the Linearity of Linguistic Relations in Language Model Embedding Spaces"
_ICLR.cc/2026/Workshop/Sci4DL — Sci4DL 2026_

### Official Review · Reviewer_11FG · 2026-02-23

**Fit:** 3
**Significance:** 2
**Confidence:** 2

**Summary:**

The paper "Analysing the Linearity of Linguistic Relations in Language Model Embedding Spaces" proposes a framework, based on Singular Value Decomposition, to quantify to what extent several types of linguistic relations are linearly encoded into the embedding space of a language model. Essentially, the hypothesis is the following: if two linguistic entities (words, phrases,...) $(t_1, t_2)$ share a particular linguistic relation, then there exist two operators $A$ and $B$ that map the corresponding embeddings $(e_1, e_2)$ onto a same third vector $x$. I.e., $Ae_1 = Be_2 = x$, or $Ae_1 - Be_2 = 0$. For any two linguistic entities $(t_x, t_y)$ that do not share this particular relation, $Ae_x - Be_y \neq 0$. The authors investigate this hypothesis for several types of linguistic relations and several language models, and find that it does (mostly) hold for some relations, but markedly less so for others.

**Strengths:**

The paper addresses a timely research question, namely a better understanding of the internal structure of language models. The focus on relations, rather than, e.g., tokens, is great. The approach is also well motivated, by encapsulating it into the hypothesis that "linearly encoded linguistic properties may be more accessible to the model's downstream computation [...] and so may disproportionately influence the model's behaviour."
Overall, this work presents a welcome step into a direction not often seen (at least by this reviewer) when it comes to "explaining" (L)LM behavior.

**Suggestions:**

- You now immediately dive into the mathematical framework in S.2. It might be worthwhile to first explain the concept of a "linguistic relation" to better sketch the context. Right now, quite a bit is left up to the reader's imagination. Additionally, a listing of the relations you consider together with some examples would be welcome as well (this would fit perfectly in an appendix).
- In the last two paragraphs of S.4 you discuss how one-to-one relations appear to be more linearly encoded than one-to-many relations. What I'm missing a bit here is some reflection on how (un)expected this result is.
- GloVe vs BERT's: any idea whether the more linear encodings of the more recent models is due to 1) more training data, 2) the model architecture or 3) a conbination of both? It would be interesting in this regard to see what happens if you trained GloVe embeddings using the same amount of training data that was used for the BERT's, and see how it affects relationship encoding.
- A linear relation means there exist 2 operators that map 2 elements of a relation (e_1, e_2) onto the same embedding. Have you looked at what this mapped-onto embedding represents, if anything, or do you simply approach it as an abstract vector?
- In the introduction, you motivate the work by hypothesizing that there might be a direct influence of the linearly encoded relations on downstream tasks, however you never come back to this. Significance could be improved by showcasing a concrete relationship between linear encoding of certain relations and model behavior. (Or, potentially further down the line, showcasing a relationship between the linear relation space and LLM biases.) I realize this is only a 4p. paper, but still some hint at future work in this direction would have been nice.
- At the start of S.4, you mention expanding the BATS dataset. Some methodological details would have been nice (could again fit in an appendix).
- The step from Eq.10 to Eq.11 isn't entirely clear to me. Right before Eq.8 you define $\mathbf{x}$ as a vector, yet if I understand correctly, in Eq.11, $((\mathbf{v}^T\mathbf{U}) \odot (\mathbf{v}^T\mathbf{U}))$ is a matrix? Same in Eq. 13; $\mathbf{Ax}$ results in a matrix, not a scalar?

Some formatting remarks/suggestions:
- Line 65: $(t_i,t_2)$ --> $(t_1,t_2)$
- Line 68: "Then, if the relation..., then..." --> one of those "then"'s can be removed
- Line 71: Missing period between the equation and "For more notational..."
- Line 74/75: "then there exists an $\mathbf{M}$" --> $\mathbf{M}_r$
- Line 83-84: "Some relations... modifications. Whereas..." --> try reformulating, as this doesn't flow well, or replace by ", whereas"
- Line 89: "the condition 1" --> "the condition expressed in Eq. 1"
- Line 91 and beyond: you seem to be using $\bar{e_{12}}$; I would suggest using $\bar{e}_{12}$ instead. Same for $\bar{t}_1$ and $\bar{t}_2$
- Line 179/180: "(Eg. 12)" --> "(eg., 12)"

---

### Official Review · Reviewer_wj8o · 2026-02-27

**Fit:** 2
**Significance:** 2
**Confidence:** 3

**Summary:**

At a high level, this paper explores how language models and word vector models represent information in their embedding space.
While prior work has shown that these models "linearly encode" certain types of information, this paper identifies and addresses two gaps.
First, distinguishing which information is encoded linearly versus non-linearly; and second, systematically studying linguistic (e.g. syntactic) relations in particular.

To this end, the paper formalizes a condition for linearly encoded relations in embeddings based on the linear algebra definition of linear relations. Concretely, a relation $r$ is linearly encoded if there exists two $(d \times d)$ matrices $L_r$ and $R_r$ which project embeddings $e_1$ and $e_2$ such that $L_r e_1 + R_r e_2 = 0$ if and only if they are related by $r$. To quantify how linearly encoded a relation is, the paper proposes an approximation that can be tractably optimized to yield the best fit for these matrices and the corresponding approximation error on the constraint.

This method is then tested on various embedding models using a dataset of 40 word-to-word linguistic relations (that is an extension of an existing dataset; introduced by the paper, and characterized by one-to-many and many-to-many relations). Based on the proposed measure of approximation error, the main findings are that (1) one-to-many and many-to-many relations are less linearly encoded than one-to-one relations; and (2) language models like RoBERTa and ModernBERT encode relations more linearly than word vectors like GloVe.
The significance of these findings is framed in terms of identifying which relational structures are most linearly accessible by models.

**Strengths:**

I quite like the overarching research question and direction of the paper.
Understanding how different relations are encoded in models, specifically in terms of linear accessibility, is a well-posed problem with meaningful research gaps that this paper addresses.

The findings are also appreciably clear and interpretable, as well as interesting in my opinion.
I like that the paper went beyond reporting results on the dataset's relation types, digging deeper and finding that one-to-many and many-to-many relations in particular play a significant role.

Lastly, I found the paper to be very well written and polished.
I appreciate the attention to detail and effort that went into communicating the main ideas both clearly and concisely.

**Suggestions:**

**Justifying the particular formalization of linear relations**
I can appreciate the mathematical formalization that went into Section 2, but it was not clear to me why this framing is justified or how it relates to prior work beyond (1) theoretical inspiration from the linear representation hypothesis and (2) the linear algebra definition of linear relations. I have a few concerns in particular:
- There is no practical justification for why this framing is used instead of another. The linear algebra definition of linear relations is arguably  only principled on a surface level, and not necessarily aligned with the motivating problem of linearly accessible representations in models. I'm not convinced that linear relations as defined in the paper correspond to the notion of linearly accessible representations.
- Related is the lack of comparison with prior works. It's not clear from the paper if this framing is novel, how it differs, what limitations does it address, what limitations does it introduce, etc. with respect to existing framings.
- To me it seems this framing might deviate non-trivially from prior notions of "linearly encoded" relations in language model embeddings. For instance, I'm not certain it's consistent with the typical notion of e.g. "e(king) - e(queen) = e(man) - e(woman)" where the difference between embeddings linearly encodes relations.

**Clarifying scope of contributions with respect to prior works**
While the paper positions it's contribution as studying relations rather than concepts, I'm not convinced that this is actually a gap in prior work. The current paper doesn't really discuss prior work or substantiate that this is a meaningful research gap and contribution.

**Ruling out potential confounding effects from approximation**
While the results regarding one-to-many and many-to-many relations are interesting, I believe they might be a spurious result arising from particular approximation that is proposed. In particular, the SVD decomposition of the $2d \times p$ related embedding pair matrix $P_r$ seems like it could conceivably underlie why approximation error is worse for x-to-many relations; especially if $p > 2d$.
This is difficult to say for certain, and I might be wrong, but I nevertheless think the possibility that this approximation is confounding results should be addressed and ideally tested empirically.

**Clarifying the effect of embedding dimensionality**
One of the key findings is that language models encode relations more linearly than GloVe. While the dimensionality of GloVe is reported (300), those of the language models is not, and is likely larger. This seems like a potential explanation for your findings that should be addressed.

**Using final layer representations**
I believe there is different works by now that suggest linear encoding of relations is more prevalent in middle rather than final layers of language models. I would suggest additional experiments or at least a brief discussion addressing this potential confound.

---

### Meta-Review · Area_Chair_KuUv · 2026-03-01

**Recommendation:** Accept

**Metareview:**

The work studies whether different linguistic relations between linguistic entities correspond to linear relationships between their corresponding embeddings. They develop a method based on constrained linear optimization and apply it to the BATS dataset and the GloVe, RoBERTa, and ModernBERT embeddings. They find varying degrees of linearity across relationships and embeddings.
The question, the methodology, and the findings are a good fit for the workshop.

---

### Decision · Program_Chairs · 2026-03-02

Accept